# Efficacy and Mechanism of *Pueraria lobata* and *Pueraria thomsonii* Polysaccharides in the Treatment of Type 2 Diabetes

**DOI:** 10.3390/nu14193926

**Published:** 2022-09-22

**Authors:** Zhujun Wang, Hui Du, Wanqian Peng, Shilin Yang, Yulin Feng, Hui Ouyang, Weifeng Zhu, Ronghua Liu

**Affiliations:** 1School of Pharmaceutical, Jiangxi University of Chinese Medicine, No. 818 Yunwan Road, Nanchang 330002, China; 2State Key Laboratory of Innovative Drug and Efficient Energy-Saving Pharmaceutical Equipment, No. 56 Yangming Road, Nanchang 330006, China; 3Key Laboratory of Modern Preparation of Chinese Medicine, Jiangxi University of Chinese Medicine, No. 818 Yunwan Road, Nanchang 330002, China

**Keywords:** *Pueraria* polysaccharides, type 2 diabetes, efficacy, metabolomics, intestinal flora, mechanism of action

## Abstract

Diabetes is called a “wasting and thirsting disorder” in Chinese traditional medicine because there is a depletion of vital substances in the body independent of the intake of food or water and an inability to reintroduce fluids through drinking. *Pueraria lobata* (Willd.) Ohwi (GG) and *Pueraria thomsonii* Benth. (FG) are traditional Chinese herbal medicines used in the treatment of wasting-thirst that reduce blood glucose levels. Flavonoids are the main pharmacodynamic components of GG and FG, and they are also the most studied components at present, but polysaccharides are also active components of GG and FG, which, however, are less studied. Therefore, this study aimed to investigate the effect of *Pueraria* polysaccharides (GG and FG polysaccharides) on type 2 diabetes (T2D), as well as their related mechanisms of action in terms of both intestinal flora and metabolomics. The C57BL/KsJ-db/db mouse model, a well-established model of obesity-induced T2D, was used in this study. The metabolomic analysis showed that *Pueraria* polysaccharides improved the metabolic profile of diabetic mice and significantly regulated metabolites and metabolic pathways. Both GG and FG polysaccharides regulated insulin resistance in mice by regulating PPAR signaling pathway so as to treat T2D. Additionally, *Pueraria* polysaccharides regulated the structure of gut microbiota and improved the diabetes-related metabolic pathway. Therefore, this study discovered the antidiabetic effects and potential mechanisms of *Pueraria* polysaccharides through multiple pathways involving gut microbiota and metabolites, providing a theoretical basis for further studies on their effects in the treatment of T2D.

## 1. Introduction

Type 2 diabetes (T2D) is a chronic metabolic disorder characterized by hyperglycemia, insulin resistance, and relative insulin deficiency, accounting for 90% of all cases of diabetes [1]. The prevalence and incidence of T2D have increased rapidly around the world in recent years. According to statistics from the 10th edition of the Global Diabetes Overview in 2021, 537 million people worldwide suffer from diabetes, and the number is expected to rise to 643 million by 2030 [2]. The etiology and mechanism of diabetes are very complicated and have not been completely clarified. It has been reported that the metabolites of T2D mice are significantly different compared to healthy mice, and the differential metabolites are involved in the glycerophospholipid metabolism pathways, the glycerol metabolism pathway [3], the arachidonic acid metabolism pathway, and the linoleic acid metabolism pathway [4]. Retinol binding protein 4 is produced by the liver and adipose tissue and plays a physiological role in the transport of vitamin A (retinol), as well as an important role in insulin resistance [5]. In addition, the intestinal microbiota, consisting of 1013–1014 bacteria, is closely related to diabetes progression [6]. The imbalance of the intestinal flora affects the metabolism of endogenous substances, such as short-chain fatty acids, bile acids, and other metabolic pathways, leading to the occurrence of diabetes [7]. A study on the intestinal flora in T2D revealed that the abundance of Firmicutes in patients with diabetes is significantly lower than in normal people [8], while the abundance of Bacteroidetes and Klebsiella are higher than those in normal people [9], thereby increasing the prevalence of insulin resistance [10]. Thus, the combination of the evaluation of gut microbiota and metabolomics might represent an effective approach to studying the pathogenesis of T2D and the mechanism of action of therapeutic drugs.

*Pueraria lobata* (Willd.) Ohwi (GG) and *Pueraria thomsonii* Benth. (FG) are traditional Chinese herbal medicines and *Chinese Pharmacopoeia* records report that they are used in the treatment of wasting-thirst, which reduces blood glucose levels [11]. *Pueraria* polysaccharides and flavonoids are the main active components; however, small flavonoid molecules are widely studied, while the effects of *Pueraria* polysaccharide macromolecules on diabetes have been less investigated. In recent years, polysaccharides in Chinese herbal medicine have attracted much attention because of their biological activities, such as anti-tumor, anti-oxidation, anti-diabetes, and antiviral effects, as well as protection against radiation, lower hyperlipidemia, and immune regulation [12]. Previous studies have shown that certain polysaccharides from traditional Chinese medicine can affect the ecological structure and metabolism of gut microbiota, thus improving the health status of the host [13]. Ganoderma lucidum polysaccharides reduce the ratio of Firmicutes to Bacteroidetes in the gut microbiota of mice fed a high-fat diet, maintain the integrity of the intestinal barrier, and reduce metabolic endotoxemia, thus reducing weight and fat accumulation in mice [14]. Laminaria japonica polysaccharide prevents high-fat-diet-induced insulin resistance in mice through the regulation of gut microbiota [15]. *Pueraria* polysaccharides have recently attracted the attention of researchers. Previous studies found that *Pueraria* polysaccharides exert a hypoglycemic effect by activating the PI3K/AKT signaling pathway [16] through the up-regulation of PI3K and AKT expressions and the down-regulation of the expressions of the FoxO1, PCK2, and G6P enzymes in insulin-resistant cells so as to improve insulin resistance in diabetes patients [17]. They also inhibit α-amylase and α-glucosidase, thus treating T2D [18]. However, in the study of XU C, et al., only GG polysaccharide was used in the treatment of T2D, and its mechanisms of action were mainly inflammatory metabolism, the PI3K/AKT signaling pathway and antioxidant activity. Thus, further studies are needed to establish whether FG polysaccharide has the same anti-diabetic effect and which mechanism of action is exerting its effect.

Both GG and FG can treat diabetes, but few studies have compared the efficacy and mechanisms of the two. Thus, a comparison of the effects of the two merits further discussion. Therefore, the db/db mouse model of T2D was used in this study, and pharmacodynamic experiments on GG and FG polysaccharides were performed, as well as investigations of their antidiabetic mechanism by using metabolomic and intestinal flora analysis, revealing the relationship between the two. Hence, this study provides the potential mechanism of action of GG and FG polysaccharides in the treatment of T2D.

## 2. Materials and Methods

### 2.1. Main Materials

#### 2.1.1. Chemicals and Drugs

Pure distilled water was purchased from Wahaha (Hangzhou, China); 2-chloro-L-phenylalanine was purchased from McLean Technologies. Acetonitrile and methanol (HPLC grade) were produced by Fisher Scientific (Fair Lawn, NJ, USA); leptin (LEP), adiponectin (ADP), glucagon-like peptide 1(GLP-1), free fatty acid (FFA), interleukin-6 (IL-6), and tumor necrosis factor (TNF-α) detection kits were purchased from Mlbio. LKB1, AMPK, p-AMPK, TSC2, p-TSC2, mTOR, p-Mtor, PPARγ, GAPDH, and β-ACTIN were purchased from CST USA.

#### 2.1.2. Medicinal Materials and Extraction

GG and FG, the dried roots of the legume *Pueraria lobata* (Willd.) Ohwi and the dried root of the legume *Pueraria thomsonii* Benth., respectively, were purchased from Zhangshu City, Jiangxi Province. GG (30 g) was dissolved in 600 mL pure water, extracted for 120 min at 100 °C two consecutive times, concentrated, and cooled down to room temperature. Next, 95% ethanol was added at a volume ratio of 1:8 to perform alcohol precipitation overnight. The mixture was filtered, centrifuged at 4000 rpm/min for 10 min, and the supernatant was dried. The difference in the extraction of FG consisted of the addition of 95% ethanol at a volume ratio of 1:2 for alcohol precipitation, while the other steps were the same as GG.

#### 2.1.3. Animals and Treatments

Forty 8-week-old specific-pathogen-free (SPF) male C57BL/KsJ-db/db mice (fasting blood glucose ≥ 11.1 mmol/L) and 10 m/m mice (C57BL/KsJ mice with normal glucose and lipid metabolism) were purchased from Cavens Co., Ltd. (Changzhou, China, License Number: SCXK (Su) 2021-0013). The mice were allowed to acclimatize for 7 days under standard laboratory conditions, with access to food and water ad libitum. Subsequently, db/db mice were randomly divided into model group, that is, db/db mice with no treatment, and three administration groups: db/db mice treated with metformin, GG polysaccharide, or FG polysaccharide. In total, 10 mice were in each group, and 10 m/m mice were in the normal group (mice with normal glucose and lipid metabolism). Metformin, GG polysaccharide, and FG polysaccharide were administered by gavage at a dose of 200 mg/kg·day for 5 weeks [19], while the normal and model groups were treated with ultrapure water at a dose of 200 mg/kg·day, also for 5 weeks. The mice were sacrificed at the end of the experiment, and serum samples, liver tissue, perivisceral fat, and cecum contents were collected and stored at −80 °C for subsequent analysis. All animal studies were approved by the Ethics Committee of the Experimental Animal Center of Jiangxi University of Traditional Chinese Medicine (Nanchang, China, SCXK 2016-0006). Animal experiments performed in this study were conducted in accordance with the Local Guide for the Care and Use of Laboratory Animals.

### 2.2. Pharmacodynamics of GG and FG Polysaccharides on T2D Mice

#### 2.2.1. Phenotypic Index Detection

The food and water intake of the mice was measured, and the body weight and fasting blood glucose concentrations were measured weekly. Then, the oral glucose tolerance test (OGTT) and insulin tolerance test (ITT) were performed after the administration period.

#### 2.2.2. Determination of the Biochemical Value, Liver Index, and Fat Index

Insulin, LEP, ADP, GLP-1, and FFA in the serum, as well as TNF-α and IL-6 in the perivisceral fat of the mice, were detected according to ELISA kit procedures. The liver index, fatty tissue index, and insulin resistance index of the mice were calculated according to liver weight, perivisceral fat weight, and fasting insulin values.

#### 2.2.3. H&E Staining

Liver tissue was soaked in tissue fixative for more than 24 h. After fixing, it was smoothed with a scalpel and put in an embedding box, soaked, and decolorized with different concentrations of ethanol, xylene, and paraffin. The dehydrated and waxed tissue was embedded with an embedding machine. Tissue sections were stained with hematoxylin and eosin. Histopathological changes were observed by electron microscopy.

### 2.3. Treatment of T2D Mice with GG and FG Polysaccharides Based on Metabonomics

#### 2.3.1. Treatment of Serum Samples

Working solution: 40 μL of serum sample was collected, the working solution (2-chloro-L-phenylalanine 10.16 μg/mL) was added in a ratio 1:4, and the mixture was vortexed for 30 s. After standing for 10 min, the mixture was centrifuged (13,000 rpm/min at 4 °C for 10 min) and the supernatant was collected and placed in the sample bottle for analysis. Quality control (QC): 10 μL of each serum sample was collected and vortexed, and then we took 100 μL from this mixture as a serum sample for processing.

#### 2.3.2. LC-MS

Ultra-high-performance liquid chromatography–quadrupole time-of-flight mass spectrometry (Waters UPLC I-Class, Waters SYNAPT G2-S) was used for LC-MS. A Waters ACQUTITY UPLCTM T3 Column (2.1 mm × 100 mm, 1.8 μm) was used at a flow rate of 0.3 mL/min, the injection volume was 1 μL, and the column temperature was 40 °C. The mobile phase consisted of 0.1% formic acid in water (A) and acetonitrile (B), and the gradient of the mobile phase B was as follows: 0–0.3 min, 5% B; 0.3–2 min, 50% B; 2–13 min, 68% B; 13–15 min, 79% B; 15–16 min, 84% B; 6–17 min, 92% B; 17–17.01 min, 5% B; and 17.01–20 min, 5% B. Mass spectrometry conditions: Electrospray ion source, full information tandem mass spectrometry, profile data, mass scanning range: 50–1200 Da, capillary voltage 3.0 kV (positive), 2.5 kV (negative), collision energy 25–45 V.

#### 2.3.3. Data Processing and Analysis

MS data were processed for peak detection, matching and alignment using Progenesis QI software (version 2.4, Waters, Milford, MA, USA). The normalized data were imported into the EZinfo software (version 3.0, Waters, USA) to select metabolites with a VIP score greater than 1 and a P value by ANOVA less than 0.05 as differential metabolites. The data file was saved and the differential metabolites were identified using the database Human Metabolome Database (http://www.hmdb.ca/) (accessed on 11 June 2021). to identify whether they were endogenous components; the mass error was 5 ppm, and the secondary fragments were recorded. PCA, PLS, and OPLS-DA were performed using SIMCA-P software (version 14.1, Umetrics, Sweden). The accuracy of OPLS-DA analysis was verified by PRT analysis. GraphPad Prism (version 8.0, Harvey Motulsky, San Diego, CA, USA) was used to draw the graphs; MeV software (version 4.9.0, TIGR, Walnut Creek, CA, USA) was used for heat mapping. MetaboAnalyst 5.0 (https://www.metaboanalyst.ca/) (accessed on 23 July 2021) online software was used to evaluate the enrichment of the metabolic pathways.

### 2.4. Effects of GG and FG Polysaccharides on Intestinal Microflora of T2D Mice

The entire intestine of the mouse was removed using a sterile scalpel, and the outer surface of the intestine was quickly cleaned in pre-cooled normal saline. The required intestinal segments were cut, and the segments were cut and opened. The content was collected and placed in an enzyme-free cryopreservation tube, stored in an ice box at −80 °C, and defrosted for sample testing. The samples were sent to Shanghai Magi Biomedical Technology Co., LTD. to perform the requested measurements. Data analysis was performed in the United States using biological cloud platforms (https://login.majorbio.com/login) (accessed on 15 August 2021). Alpha diversity analysis was performed using Chao 1, Ace, Shannon, and Simpson indices. ANOSIM analysis was used to analyze the similarity between groups, and LEfSe analysis was used to analyze the genus level differences between the normal group and model group.

### 2.5. Western Blotting

Liver samples were homogenized in tissue lysis buffer (RIPA: protease inhibitor: phosphorylated protease inhibitor = 100:1:1) and centrifuged at 12,000 rpm for 15 min at 4 °C. The supernatant was collected and stored at −80 °C for later use. The protein concentration of the sample was measured according to the BCA kit procedure. Proteins were separated by electrophoresis and then transferred to a polyvinylidene fluoride membrane, which was incubated at 4 °C overnight with specific antibodies (LKB1, AMPK, p-AMPK, TSC2, p-TSC2, mTOR, p-Mtor, PPARγ, GAPDH, β-ACTIN) (CST, United States), followed by incubation with secondary antibodies (anti-rabbit IgG, HRP-linked antibody) (CST, Danvers, MA, USA) for 1.5 h at room temperature. Next, the membrane was washed three times with 1 × TBST solution for 10 min each time. The membranes were submerged in the developer solution after absorbing the water with filter paper and then placed in a gel imaging scanning system for development. The intensity of the gray bands was quantified using Image Lab software (version 4.0, Bio-Rad, Hercules, CA, USA).

### 2.6. Statistical Analysis

Statistical analysis was performed using IBM SPSS Statistics (version 21.0, SPSS, Armonk, NY, USA) while graphs were drawn using Graph Pad Prism (version 8.0, Harvey Motulsky, USA). Comparison between groups was performed by one-way ANOVA, and the results were expressed as mean ± standard deviation ($\bar{x}\pm s$). A value of *p* < 0.05 was considered statistically significant.

## 3. Results

### 3.1. Pharmacodynamic Analysis

#### 3.1.1. Effects of Pueraria Polysaccharides on Physiological Indexes

The weekly body weight, food intake, water intake, and fasting blood glucose of the mice were examined for the entire 5 weeks of the experiment (Figure 1A–D), and the results showed that GG and FG polysaccharides alleviated the symptoms of “three more and one less” (food intake and water intake, fasting blood glucose, weight loss) in type 2 diabetic mice. Glucose tolerance (Figure 2) showed that GG polysaccharides significantly enhanced the glucose metabolism in the mice, and FG polysaccharides had a tendency to increase the glucose metabolism in the mice. In addition, GG and FG polysaccharides improved insulin sensitivity in the mice (Figure 3). No statistically significant difference was observed between the effects of GG polysaccharides and FG polysaccharides.

#### 3.1.2. Biochemical Indexes and Liver Histopathological Analysis

Serum biochemical indices (Table 1) showed that insulin, ADP, and GLP-1 were up-regulated, while LEP, FFA, IL-6, and TNF-α decreased. In addition, a decreasing trend in the insulin resistance index, liver index, and adipose index after *Pueraria* polysaccharide administration was observed. No significant difference in the above biochemical indexes between the GG polysaccharide and FG polysaccharide groups was observed. Moreover, the alveolar fat droplets decreased, and nuclear pyknosis and apoptosis improved to varying degrees after the intervention of metformin, GG polysaccharide, and FG polysaccharide, as shown in Figure 4.

### 3.2. Metabolomic Analysis

#### 3.2.1. Endogenous Metabolites and Metabolic Pathways Associated with T2D Mice

PCA (Figure 5A,B) and OPLS-DA(ESI-: R^2^Y-0.994, Q^2^-0.971; ESI+: R^2^Y-0.938, Q^2^-0.752) (Figure 5C,D) both indicated differences in serum metabolites between the model group and the normal group. Furthermore, the 200 permutations test (Figure 6) showed that the OPLS-DA model had a high explanatory power and predictive ability. EZinfo software was used to identify 67 differential metabolites (Table 2) and the heat map (Figure 7) showed that the normal group and the model group were clearly divided into two categories. The enriched metabolic pathways where the different metabolites were involved are shown in Figure 8. The topological analysis of the metabolic pathways showed that the different metabolic profiles between T2D and normal mice were mainly due to the abnormal metabolism of unsaturated fatty acid biosynthesis, glycerophospholipid, arachidonic acid, α-linolenic acid, glycerol, retinol, and steroid biohormone.

#### 3.2.2. Metabolomics Analysis of Serum after GG and FG Polysaccharide Treatment in T2D Mice

The PCA (Figure 9A,B and Figure 10A,B) and PLS-DA (Figure 9C,D and Figure 10C,D) showed that the GG polysaccharide group and FG polysaccharide group were located between the normal group and the model group. The results showed that the metabolic profile and changes in the metabolites of the model group were closer to those in the normal group after intervention with GG polysaccharides and FG polysaccharides, indicating that they ameliorated T2D, which was consistent with the results of the pharmacodynamic experiment. Figure 11A shows the 10 metabolites with evident recall after the administration of GG polysaccharide; a significant increase in PE-NMe2 (18:0/20:4 (8Z,11Z,14Z,17Z)), PGP (18:0/PGF1alpha), PC (20:2(11Z,14Z)/TXB2), N-oleoyl phenylalanine, and PS (18:1(11Z)/6 keto-PGF1alpha) was observed, as well as a reduction in tauroursocholic acid, 13-HODE, lysoPC (20:4(5Z,8Z,11Z,14Z)/0:0), alpha-linolenic acid, and PA (16:0/18:1(11Z)). These metabolites were involved in the biosynthesis of unsaturated fatty acids, glycerol phospholipid metabolism, and α-linolenic acid metabolism. Figure 11B shows the four metabolites with evident recall after the administration of FG polysaccharide; a significant increase in PC (PGF2alpha/2:0), lysoPE (20:1(11Z)/0:0), and lysoPC (16:1(9Z)/0:0) was observed, as well as a reduction in uric acid. The main pathway involved in these metabolites was glycerophospholipid metabolism.

#### 3.2.3. Correlation of Differential Metabolites and Biochemical Indices

The Spearman correlation analysis was carried out on the differential serum metabolites and biochemical indicators. The results shown in Figure 12A,B, showed that the absolute value of the correlation coefficient between the tauroursocholic acid and all the biochemical indexes was the largest in the GG polysaccharide group, while the absolute value of the correlation coefficient between the uric acid and all the biochemical indexes was the largest in the FG polysaccharide group, indicating that GG polysaccharides significantly decreased the level of tauroursocholic acid in the treatment of T2D. FG polysaccharide significantly decreased the uric acid level in the treatment of T2D.

### 3.3. Gut Microbiota Analysis

#### 3.3.1. Effects of Pueraria Polysaccharide on the Species Composition of Gut Microbiota

According to the Chao 1, Ace, Shannon, and Simpson index analyses (Table 3), the cecal microflora was richer and more diverse after *Pueraria* polysaccharide intervention. The coverage index of each group was above 0.99, indicating that the sequencing results represented the real situation of the microorganisms in the sample. There was no significant difference in the above parameters between the GG polysaccharide and FG polysaccharide groups.

The analysis of the phylum (Figure 13A) revealed a higher relative abundance of Bacteroidetes, as well as lower levels of Firmicutes, desulphurobacteria, Campylobacter, Deferrobacteriaceae, and actinobacteria in the model group than in the normal group. *Pueraria* polysaccharide treatment reversed the relative abundance of these bacteria modified by T2D, almost reaching the same situation in the normal mice, as shown in Figure 13B. No significant difference between the effects of the GG polysaccharides and FG polysaccharides was observed.

The analysis of the genus shown in Figure 14A with the PCoA analysis of all samples based on the weighted-unifrac distance algorithm revealed a clear separation and evident spatial clustering of the plots. The boxplots in Figure 14B represent the distribution of different groups of samples on the PC1 axis, revealing that the difference between the normal group and model group was the largest, and the distance between the GG polysaccharide group and model group was larger than the distance between the FG polysaccharide group and model group. The ANOSIM analysis demonstrated significant differences among the samples, but no significant difference was observed between the GG and FG polysaccharide groups. LEfSe analysis was used with an LDA threshold of two to analyze the differences in the genus levels between the normal group and the model group, and the multi-group comparison strategy was one-against-all. A significant difference in bacterial abundance at the level of 45 genera in the model group was observed, as shown in Figure 15A. The bacterial abundance of 18 genera was significantly affected (*p* < 0.05) after 5 weeks of FG polysaccharide intervention, and the bacterial abundance of 14 genera was significantly affected (*p* < 0.05) after 5 weeks of GG polysaccharide intervention, as shown in Figure 15B,C. A total of 20 bacteria had a callback trend at the genus level after the intervention of FG polysaccharide (Figure 16A), and among them, g_Helicobacter, g_norank_f_Ruminococcaceae, and g_Colidextribacter were significantly up-regulated, while the g_Eubacterium_siraeum_group and g_Klebsiella significantly decreased. A total of 23 bacteria had a callback trend at the genus level (Figure 16B), and among them, g_Helicobacter, g_Romboutsia, and g_UBA1819 were significantly up-regulated, while g_Parasutterella, g_Faecalibaculum, and g_Weissella significantly decreased. GG polysaccharides and FG polysaccharides significantly regulated the content of different bacteria except for g_Helicobacter.

#### 3.3.2. KEGG Enrichment Analysis

Three levels of information and abundance tables of metabolic pathways were obtained via PICRUSt function prediction. In pathway level 1, the expression of metabolism, organic systems, genetic information processing, environmental information processing, cellular processes, and human disease pathways in the model group was lower than that of the normal group. A callback trend was observed after the intervention with *Pueraria* polysaccharide (Figure 17). A total of 46 metabolic pathway subfunctions were enriched at pathway level 2; these metabolism-related pathways consisted of carbohydrate metabolism, amino acid metabolism, energy metabolism, lipid metabolism, and other metabolic pathways. The prediction analysis of pathway level 3 showed the involvement of 338 metabolic pathways in total, and among them, 153 metabolic pathways were significantly different between the normal group and the model group (*p* < 0.05). The analysis of these 153 metabolic pathways with significant differences revealed that they were related to diabetes, as shown in Figure 18. After *Pueraria* polysaccharide intervention reversed the abundance of these metabolic pathways. There was no significant difference between the effect of GG polysaccharide and FG polysaccharide.

### 3.4. Correlation of Differential Metabolites and Gut Microbiota 

Spearman correlation analysis was carried out on the bacterial taxa and differential serum metabolites relevant to *Pueraria* polysaccharide to explore the potential functional relationship between gut microbiota and differential metabolites. Six bacteria had a significant relationship with at least three metabolites (*p* < 0.05) after GG polysaccharide intervention; Romboutsia bacteria played a key role in reducing the level of tauroursocholic acid after GG polysaccharide treatment, as shown in Figure 19A. Five bacteria had a significant relationship with at least one metabolite (*p* < 0.05) after the intervention of FG polysaccharide; Klebsiella bacteria played a key role in reducing the level of uric acid after FG polysaccharide treatment, as shown in Figure 19B.

### 3.5. Effects of Pueraria Polysaccharide on the PPAR Signaling Pathway in T2D Mice

The protein expression of LKB1, P-AMPK, P-TSC2, and PPARγ in the liver protein of the model group decreased, and the expression of p-mTOR protein increased compared with the normal group. The protein expression of LKB1, P-AMPK, P-TSC2, and PPARγ in the liver protein of the model group increased, and the protein expression of P-mTOR decreased after *Pueraria* polysaccharide intervention compared with the normal group. No significant difference in protein expression was observed between the effect of the GG and FG polysaccharides (Figure 20).

## 4. Discussion

GG and FG have the ability to raise body fluids and quench thirst according to the *Chinese Pharmacopoeia* [11]. The hypoglycemic effect of the flavonoid extracts of these plants has been extensively studied. In addition, previous studies showed that GG polysaccharide treats diabetes [16,17,18], but the therapeutic effect of FG polysaccharide has not been reported. In this study, GG polysaccharides and FG polysaccharides were used for 5 weeks on T2D mice to evaluate their efficacy and explore their potential anti-diabetic mechanisms from the perspectives of metabolomics and intestinal flora.

The typical symptoms of T2D are polydipsia, polyuria, hypereating, weight loss, and fatigue [20]. Our results showed a slower weight gain and higher water and food intake in the model group than in the normal group, which is consistent with the above description. Nevertheless, these changes were clearly reversed after the treatment with GG polysaccharide and FG polysaccharide. Moreover, biochemical indicators such as insulin, insulin resistance index, ADP, GLP-1, LEP, FFA, IL-6, and TNF-α in the serum indicated that GG polysaccharide and FG polysaccharide improved insulin resistance and leptin resistance in diabetic mice, as well as chronic inflammation. Previous studies showed that the thirst-quenching effect of GG is associated with improved insulin resistance [21]. Our pharmacodynamic study also proved that both the GG and FG polysaccharides improved insulin resistance, indicating that both of them have the effect of increasing fluids and quenching thirst. In addition, the liver index and adipose index were subjected to a downward trend after the intervention; the liver tissue slices showed that the vesicular fat droplets in the hepatocytes were reduced, and the pyknosis and apoptosis of cells were improved to varying degrees, indicating that both polysaccharides also had a certain protective effect on the liver. The effect of GG polysaccharide in increasing fluid and quenching thirst was better than that of FG polysaccharide at the same dose according to the biochemical indexes, although no statistically significant difference was observed between the two polysaccharides.

Metabolomics techniques can be used to discover metabolites that differ between people with diabetes and healthy people, as well as the changes in metabolites after drug intervention, which is of great help in the analysis of diabetes pathogenesis and the mechanism of action in drugs. For example, a metabolomic study found that 50 biomarkers of metabolic disorders show a trend of regression after treatment with red ginseng extract, indicating that T2D can be ameliorated by improving metabolic disorders [22]. A previous article reported differential endogenous metabolites in the serum and urine samples caused by metabolomics, as well as the antidiabetic and antioxidant effects of ginsenoside reevaluated using KEGG enrichment analysis [23]. In our work, the metabolic profiles of the serum in T2D mice and normal mice were analyzed, and a total of 67 differential metabolites were found, mainly lipids, glycerol phospholipids, fatty acids, and amino acid derivatives. The pathway enrichment analysis revealed that the differential metabolites were involved in unsaturated fatty acid biosynthesis and metabolism, glycerophospholipid metabolism, arachidonic acid metabolism, α-linolenic acid metabolism, glycerolipid metabolism, retinol metabolism, and steroid biohormone synthesis. This result was consistent with the findings of previous studies on abnormal metabolic pathways in T2D [3,4,5]. The metabolic profile of T2D mice was almost back to that of the normal group after the intervention of GG and FG polysaccharides. The GG and FG polysaccharides significantly regulated some endogenous components, and both significantly regulated the glycerophospholipid metabolic pathway. The PPAR signaling pathway is an important pathway regulating lipid metabolism and is also an upstream signaling pathway for glycerophospholipid metabolism, suggesting that the two polysaccharides might exert therapeutic effects on T2D by regulating the PPAR signaling pathway. Spearman correlation analysis showed that taurouronic acid and uric acid were the key metabolites decreased by the GG and FG polysaccharides in the treatment of T2D mellitus; thus, they could be used as potential biomarkers.

16SrRNA high-throughput sequencing was performed on the bacterial V3–V4 region in the mouse cecal content to better understand the hypoglycemic mechanisms of the GG and FG polysaccharides. Alpha diversity and beta diversity analysis showed a significant difference in the intestinal flora between the normal group and the model group. However, the richness and diversity were both improved in the model group after the intervention of GG and FG polysaccharides, and the composition of the intestinal flora was almost back to that of the normal group. No significant difference in any index was observed between the effect of GG polysaccharide and FG polysaccharide. The species composition in each group was analyzed at the phylum level, and the abundance of firmicutes in the intestinal microbiota of the T2D mice was lower than in the normal group, while the abundance of Bacteroides was higher than in the normal group, which was consistent with another study revealing that the abundance of firmicutes in the intestinal microbiota of T2D patients was significantly lower than that of normal controls [8]. The abundance of Bacteroidetes and Firmicutes was ameliorated after GG and FG polysaccharide intervention. A difference in 45 bacterial genera was found between the model group and the normal group. GG polysaccharide exerted a significant callback to six species, and FG polysaccharide exerted a significant callback to five species. The pathways involved in the microbiota affected by T2D mellitus were the same as those involved in the metabolites affected by T2D mellitus in serum metabolomics. The significantly increased Romboutsia bacteria content after GG polysaccharide treatment was negatively correlated with tauroursocholic acid, The significantly reduced Klebsiella content after FG polysaccharide treatment was positively correlated with uric acid, Romboutsia [24], and Klebsiella [10], which are qualities associated with T2D. These results indicated that T2D affected the metabolic environment in the body by affecting the gut microbiota, and the GG and FG polysaccharides were able to ameliorate the metabolites in the serum by regulating the gut microbiota. In addition, the abundance of tauroursocholic acid and uric acid related to the PPAR signaling pathway was reversed after the intervention of the GG and FG polysaccharides, suggesting that these two polysaccharides exerted a regulatory effect on the PPAR signaling pathway, which was consistent with the metabolomics results.

The results of pharmacodynamics, metabolomics, and intestinal microbiota studies showed that the effects of the GG and FG polysaccharides in the treatment of T2D mellitus were related to the regulation of the PPAR signaling pathway. The mechanism of action in the GG and FG polysaccharides was represented by the increase in the expression of LKB1 protein, P-AMPK protein, P-TSC2 protein, and PPARγ protein and the reduction in p-mTOR protein expression. These results suggested that GG and FG polysaccharides might play a role in promoting fluids and quenching thirst through the regulation of the PPAR signaling pathway.

Our results showed no significant difference in the efficacy of GG polysaccharide and FG polysaccharide in the treatment of T2D mellitus. The combination of the results of the metabolomics and intestinal microbiota analysis revealed that they improved fluids and quenched thirst using different mechanisms: GG polysaccharide reduced the concentration of taurocholic acid in the serum by increasing the abundance of Romboutsia bacteria, so as to regulate the PPAR signaling pathway and have a therapeutic effect on insulin resistance; FG polysaccharide reduced the uric acid level in the serum by reducing the abundance of Klebsiella bacteria and then regulated the PPAR signaling pathway to exert a therapeutic effect on insulin resistance. Although the regulatory pathways of the two were different, they exerted the same effect of increasing fluid and quenching thirst by regulating the same (PPAR) signaling pathway.

## 5. Conclusions

This study evaluated the effect of GG and FG polysaccharides in increasing body fluids and quenching thirst to alleviate the symptoms of “three more and one less” in T2D mice. The results of the metabolomics and intestinal flora showed that the GG and FG polysaccharides had different effects on the regulation of serum metabolites and bacteria, but both significantly regulated the glycerophospholipid metabolic pathway. The PPAR signaling pathway was also regulated by the GG and FG polysaccharides. Therefore, GG and FG polysaccharides might play a role in the treatment of T2D mellitus by regulating the PPAR signaling pathway.

## Figures and Tables

**Figure 1 nutrients-14-03926-f001:**
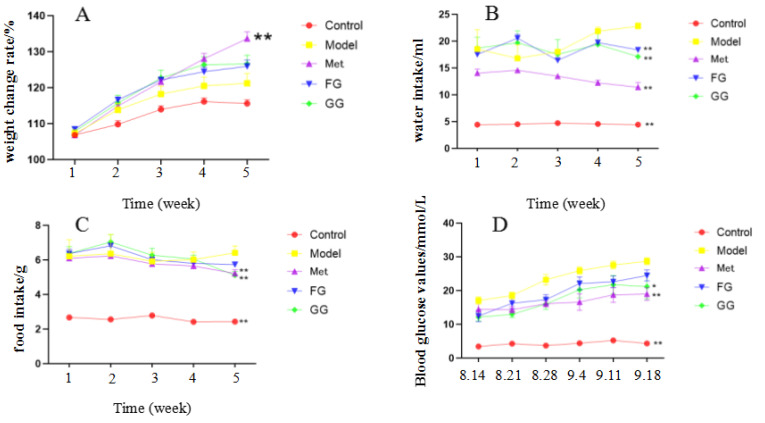
(**A**) Effect of *Pueraria* polysaccharide on body weight of db/db mice (*n* = 10; ** *p* < 0.01 compared with the model group). (**B**) Effect of *Pueraria* polysaccharide on the water intake of db/db mice (*n* = 10; ** *p* < 0.01 compared with the model group). (**C**) Effect of *Pueraria* polysaccharide on food intake of db/db mice (*n* = 10; ** *p* < 0.01 compared with the model group). (**D**) Effect of *Pueraria* polysaccharide on fasting blood glucose in db/db mice (*n* = 10; * *p* < 0.05; ** *p* < 0.01 compared with the model group).

**Figure 2 nutrients-14-03926-f002:**
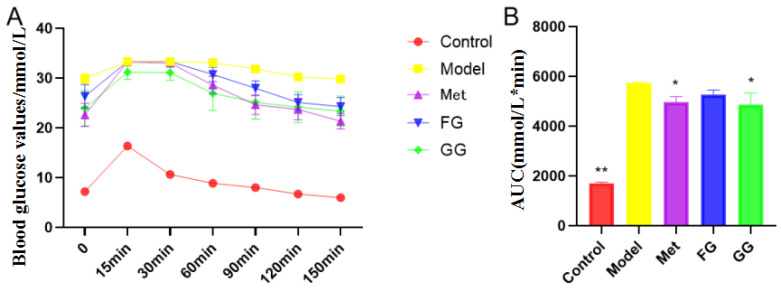
Effects of *Pueraria* polysaccharide on the glucose tolerance of db/db mice (*n* = 10; * *p* < 0.05; ** *p* < 0.01 compared with the model group). (**A**) Effect of *Pueraria* polysaccharide on blood glucose value in db/db mice; (**B**) Effect of *Pueraria* polysaccharide on AUC in db/db mice.

**Figure 3 nutrients-14-03926-f003:**
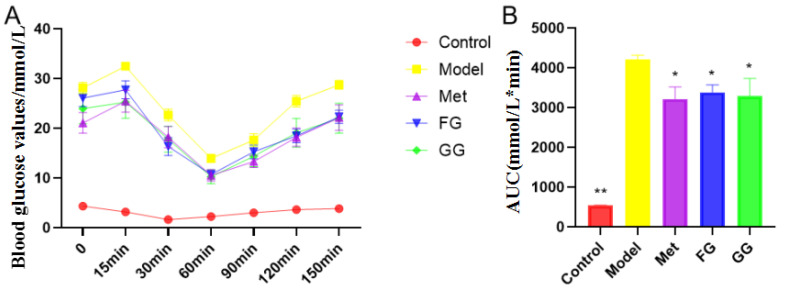
Effect of *Pueraria* polysaccharide on the insulin tolerance of db/db mice (*n* = 10; * *p* < 0.05; ** *p* < 0.01 compared with the model group). (**A**) Effect of *Pueraria* polysaccharide on blood glucose value in db/db mice; (**B**) Effect of *Pueraria* polysaccharide on AUC in db/db mice.

**Figure 4 nutrients-14-03926-f004:**
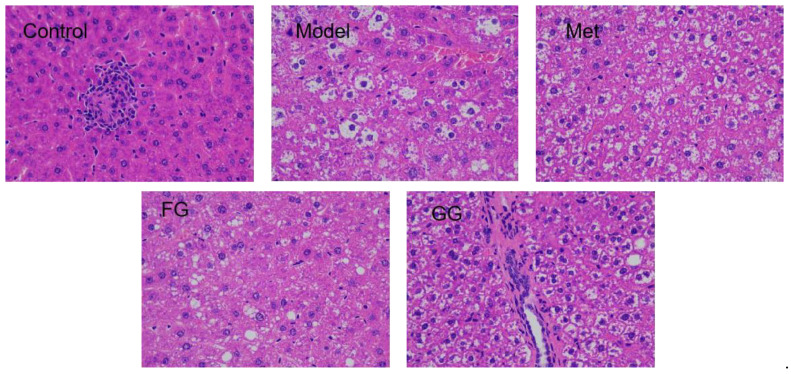
H&E staining of the liver tissue of db/db mice (400× magnification).

**Figure 5 nutrients-14-03926-f005:**
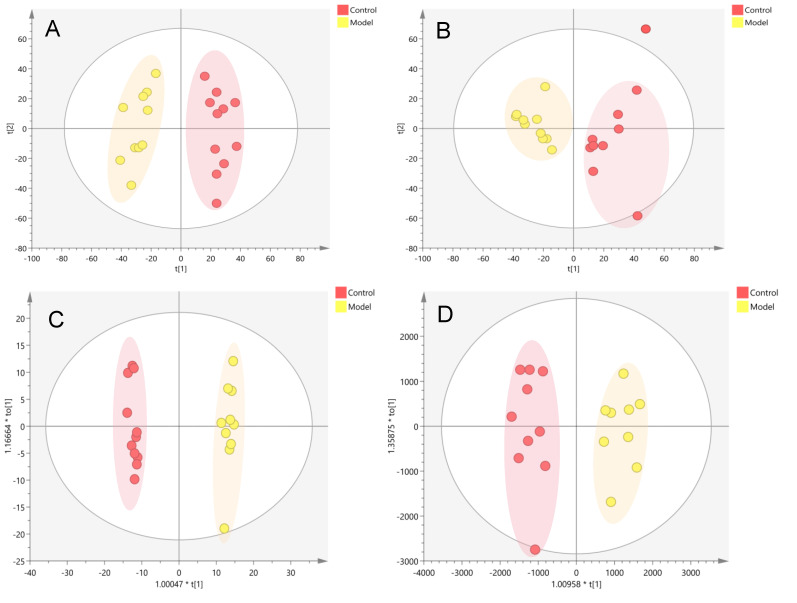
Metabolic profile analysis of the serum of the T2D model in db/db mice ((**A**). NEG, PCA; (**B**). POS, PCA; (**C**). NEG, OPLS-DA; (**D**). POS, OPLS-DA).

**Figure 6 nutrients-14-03926-f006:**
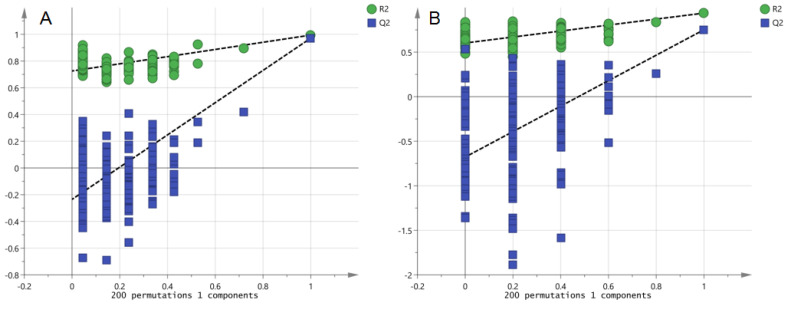
PRT analysis of the T2D model in db/db mice ((**A**). NEG, PRT; (**B**). POS, PRT).

**Figure 7 nutrients-14-03926-f007:**
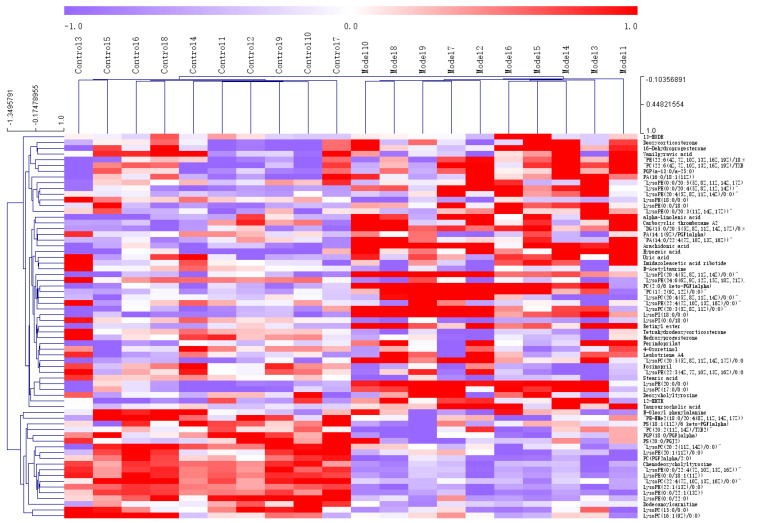
Heat map analysis of differential metabolites in T2D db/db mice. (Control 1–10 represents Table 1. Model represents the model group).

**Figure 8 nutrients-14-03926-f008:**
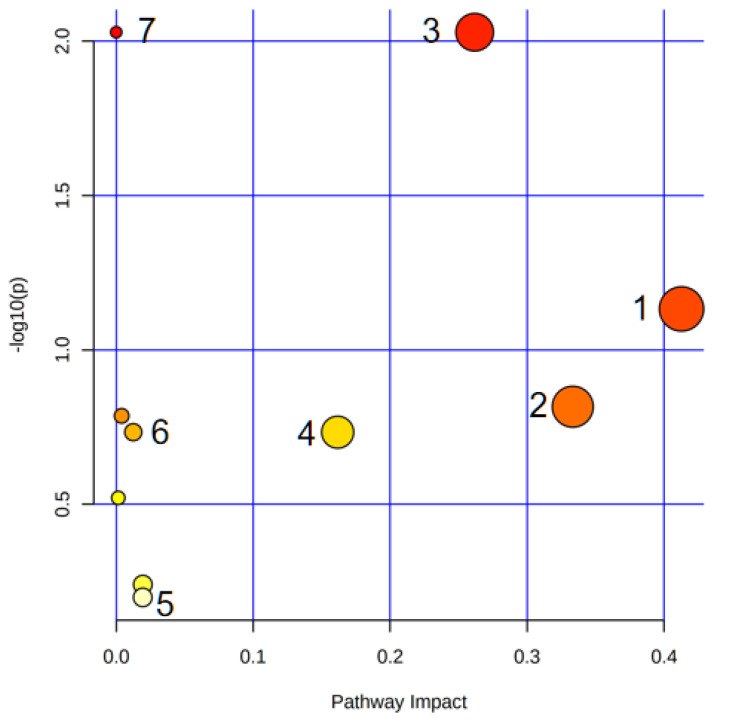
Overview of the metabolic pathway analysis. 1. Arachidonic acid metabolism; 2. Alpha-linolenic acid metabolism; 3. Glycerophospholipid metabolism; 4. Retinol metabolism; 5. Steroid hormone biosynthesis; 6. Glycerolipid metabolism; 7. Biosynthesis of unsaturated fatty acids.

**Figure 9 nutrients-14-03926-f009:**
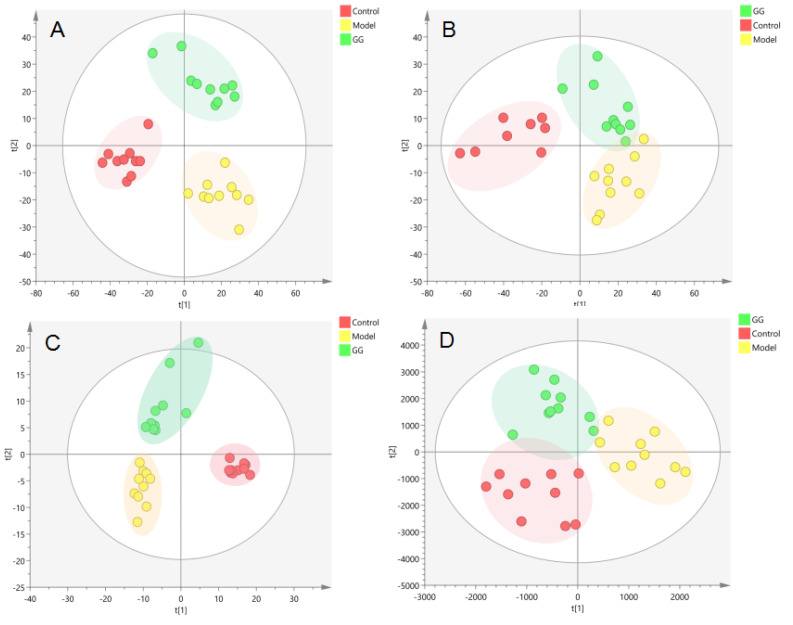
Metabolic profile in the serum of the T2D db/db mice treated with GG polysaccharide ((**A**). NEG, PCA; (**B**). POS, PCA; (**C**). NEG, PLS-DA; (**D**). POS, PLS-DA).

**Figure 10 nutrients-14-03926-f010:**
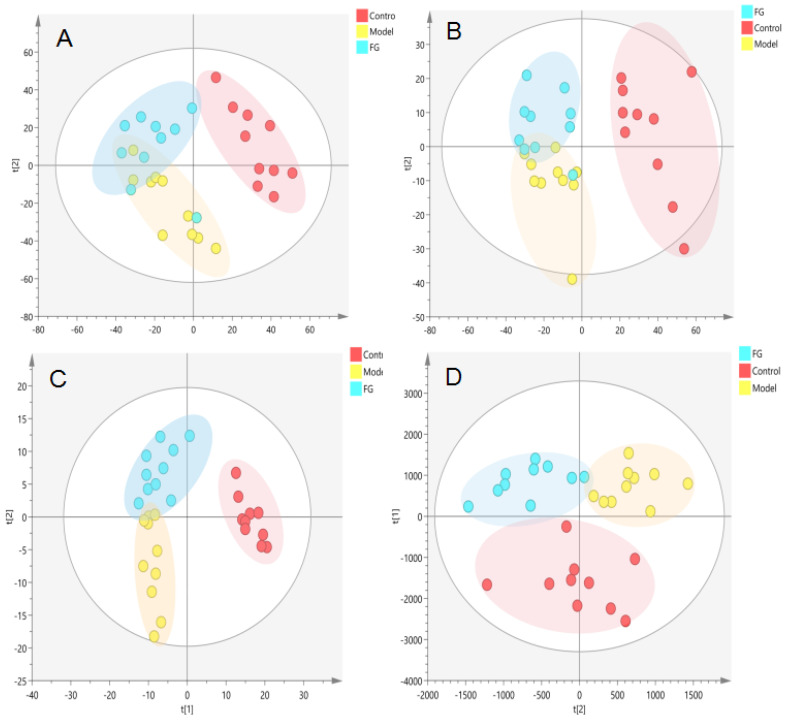
Metabolic profile in the serum samples of T2D db/db mice treated with FG polysaccharide ((**A**). NEG, PCA; (**B**). POS, PCA; (**C**). NEG, PLS-DA; (**D**). POS, PLS-DA).

**Figure 11 nutrients-14-03926-f011:**
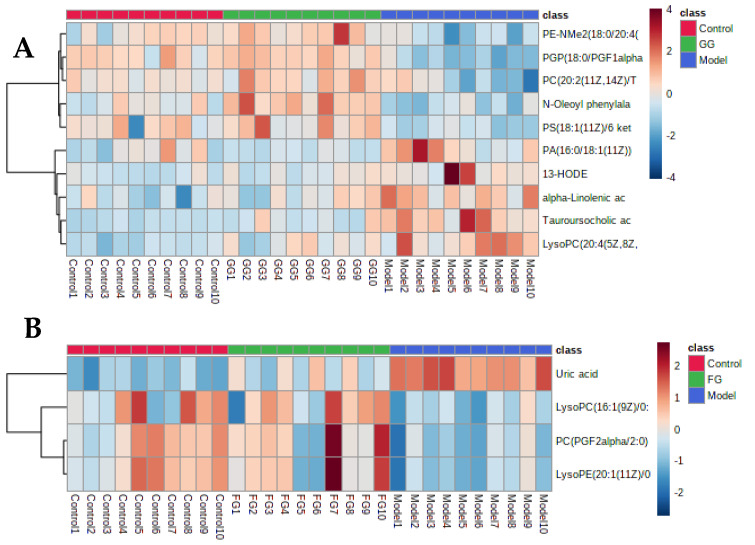
(**A**) Heat map analysis of differential metabolites in the serum after GG polysaccharide treatment. (**B**) Heat map analysis of differential metabolites in the serum after FG polysaccharide treatment.

**Figure 12 nutrients-14-03926-f012:**
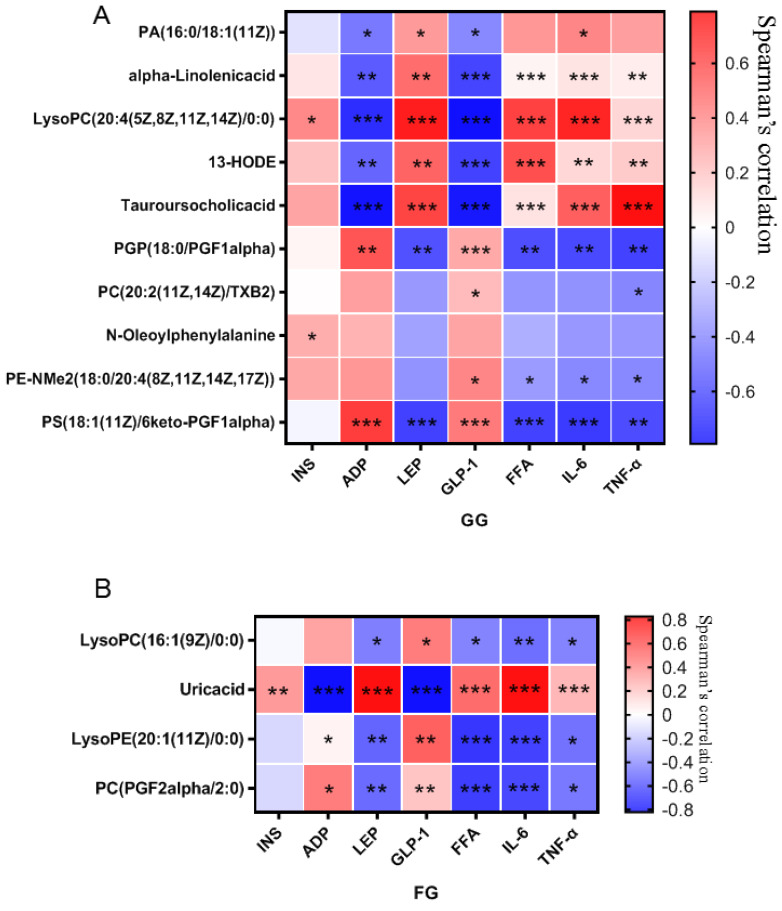
Correlation analysis of metabolites and biochemical indices affected by *Pueraria* polysaccharide intervention ((**A**): GG; (**B**): FG; red represents a positive correlation, blue represents a negative correlation, * *p* < 0.05, ** *p* < 0.01, *** *p* < 0.001).

**Figure 13 nutrients-14-03926-f013:**
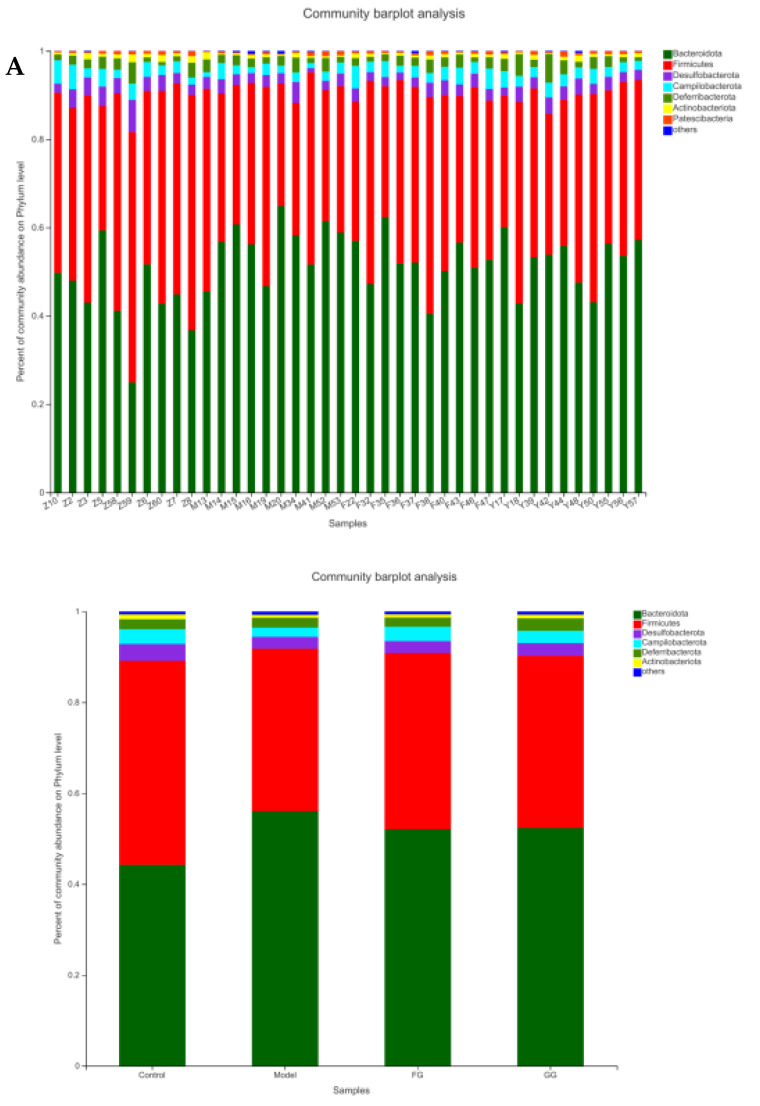
Changes in the relative abundance of cecal microflora at the phylum level in db/db mice (*n* = 10). (**A**) community barplot analysis on Phylum level; (**B**) community heatmap analysis on Phylum level.

**Figure 14 nutrients-14-03926-f014:**
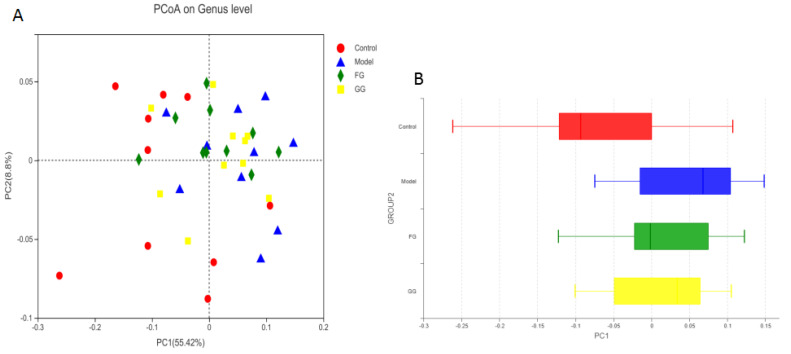
(**A**) PCoA analysis of the cecal microflora in db/db mice based on the weighted-unifrac distance (*n* = 10); (**B**) the distribution of different groups of samples on the PC1 axis.

**Figure 15 nutrients-14-03926-f015:**
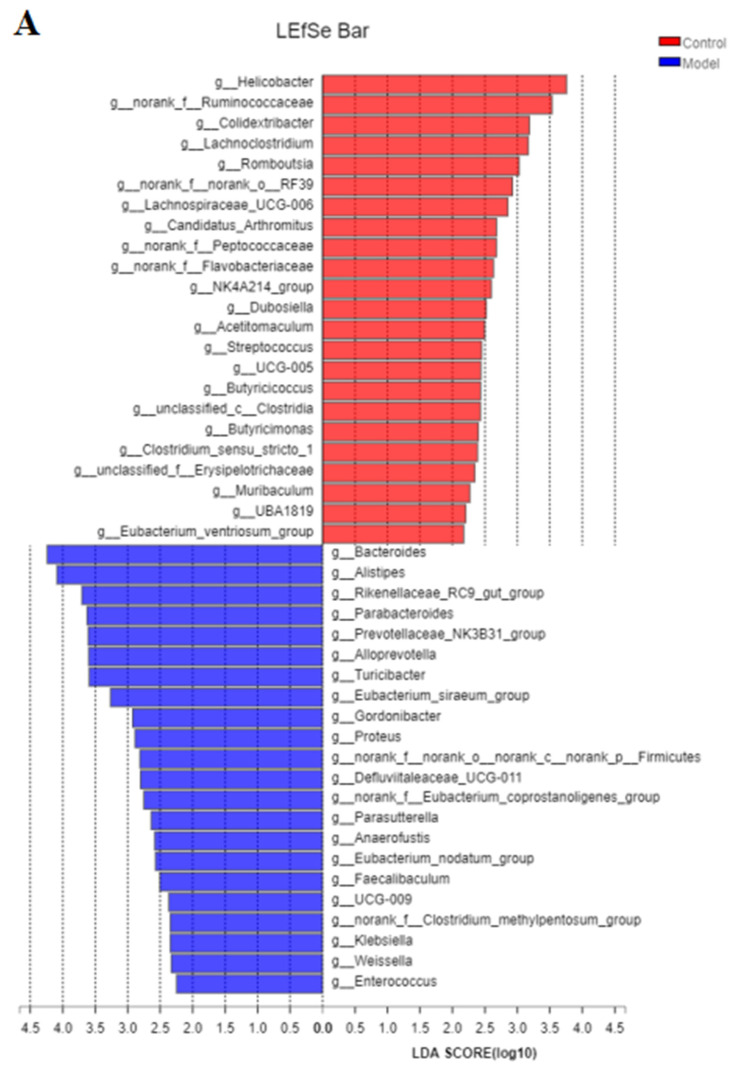
LEfSe linear discriminant analysis of the cecal microflora in db/db mice ((**A**). normal group vs. model group; (**B**). model group vs. FG group; (**C**). model group vs. GG group).

**Figure 16 nutrients-14-03926-f016:**
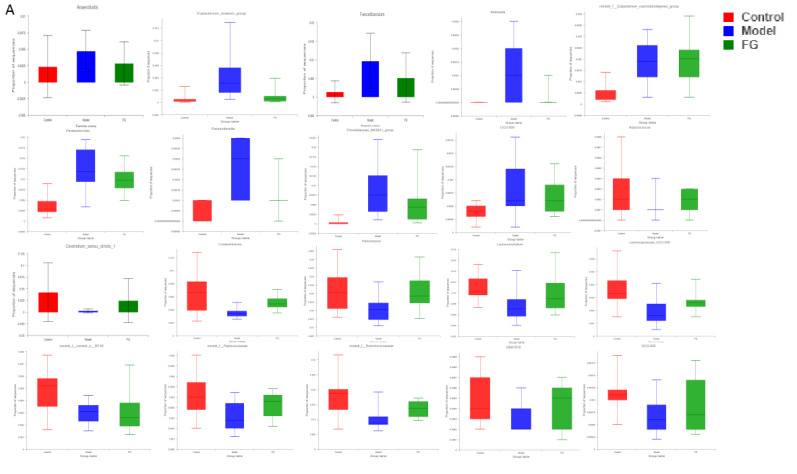
Effects of *Pueraria* polysaccharide on the cecal microflora in db/db mice ((**A**). FG; (**B**). GG).

**Figure 17 nutrients-14-03926-f017:**
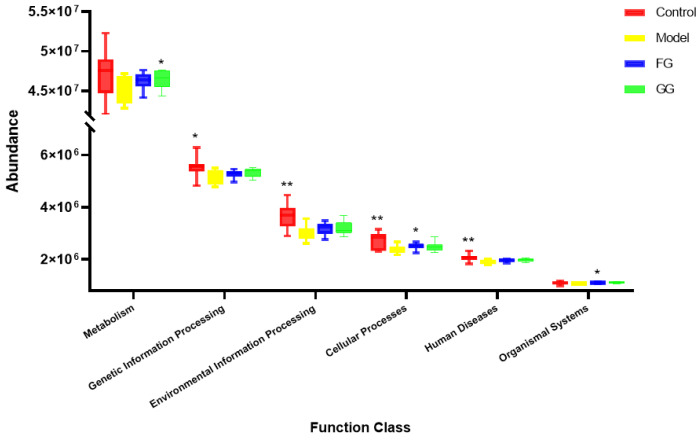
KEGG pathway level 1 box diagram (*n* = 10; * *p* < 0.05; ** *p* < 0.01 compared with the model group).

**Figure 18 nutrients-14-03926-f018:**
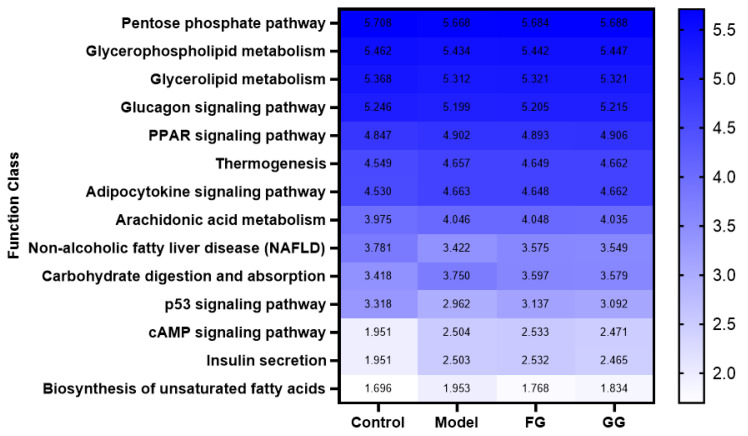
Heat map of KEGG pathway level 3 functional abundance (*n* = 10).

**Figure 19 nutrients-14-03926-f019:**
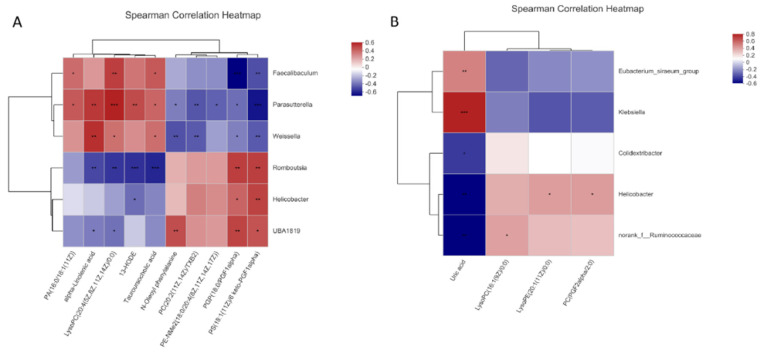
Correlation analysis between the cecal microflora and metabolites affected by *Pueraria* polysaccharide intervention ((**A**): GG; (**B**): FG; red represents a positive correlation, blue represents a negative correlation, * *p* < 0.05, ** *p* < 0.01, *** *p* < 0.001).

**Figure 20 nutrients-14-03926-f020:**
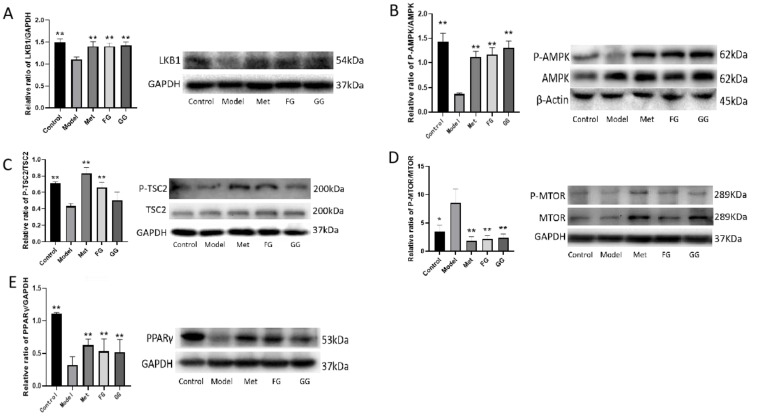
Effects of GG polysaccharide and FG polysaccharide on liver protein expression in db/db mice ((**A**). LKB1; (**B**). P-AMPK/AMPK; (**C**). P-TSC2/TSC2; (**D**). P-mTOR/mTOR; (**E**). PPAR gamma. * *p* < 0.05, ** *p* < 0.01 compared with the model group).

**Table 1 nutrients-14-03926-t001:** Effects of *Pueraria* polysaccharides on the biochemical values of db/db.

Group	Control	Model	Met	FG	GG
INS (ng/mL)	0.96 ± 0.13 *	2.19 ± 0.16	3.82 ± 0.61 **	2.56 ± 0.30	3.27 ± 0.22 *
HOMA-IR (ng/mL × mmol/L)	0.27 ± 0.10 *	3.29 ± 0.60	2.74 ± 0.78	2.63 ± 0.76	2.64 ± 1.20
ADP (μg/L)	187.52 ± 2.84 **	138.65 ± 2.26	174.46 ± 2.71 **	163.76 ± 1.85 **	163.88 ± 2.00 **
LEP (pg/mL)	860.45 ± 14.48 **	1184.66 ± 10.96	935.61 ± 14.41 **	1006.35 ± 11.18 **	1031 ± 8.01 **
GLP-1 (pmol/L)	5.77 ± 0.05 **	3.89 ± 0.07	5.32 ± 0.07 **	4.75 ± 0.05 **	4.95 ± 0.05 **
FFA (μmol/L)	782.73 ± 14.57 **	1001.58 ± 10.66	865.76 ± 8.19 **	898.70 ± 9.87 **	886.20 ± 11.54 **
IL-6 (pg/mL)	143.86 ± 2.32 **	194.81 ± 0.99	157.28 ± 2.02 **	169.20 ± 1.88 **	166.03 ± 1.42 **
TNF-α (ng/L)	795.71 ± 12.63 **	1036.50 ± 10.05	885.73 ± 10.05 **	920.54 ± 12.49**	904.75 ± 13.16 **
Liver index (%)	0.042 ± 0.005 **	0.051 ± 0.004	0.051 ± 0.003	0.049 ± 0.003	0.046 ± 0.008 *
Fatty tissue index (g/g)	0.03 ± 0.005 **	0.10 ± 0.013	0.08 ± 0.014 **	0.09 ± 0.015 *	0.08 ± 0.007 **

*n* = 10; * *p* < 0.05; ** *p* < 0. 01 compared with the model group). Abbreviations: INS—insulin; HOMA-IR—homeostatic model assessment of insulin resistance; ADP—adiponectin; LEP—leptin; GLP-1—glucagon-like peptide 1; FFA—free fatty acid; IL-6—interleukin-6; TNF-α—tumor necrosis factor.

**Table 2 nutrients-14-03926-t002:** Identification of differential metabolites in mouse serum between normal group and model group.

NO	Retention Time	Name	Formula	Experimental Mass	Ion Mode	Mass Error	MS/MS	Levels
P1	17.79	PA (16:0/18:1(11Z))	C_37_H_71_O_8_P	674.4886	M+H	0.82	661.477	↑
P2	15.95	Alpha-linolenic acid	C_18_H_30_O_2_	278.2245	M+H	−4.56	191.1780, 125.0972	↑
P3	12.23	LysoPE (0:0/20:3(11Z,14Z,17Z))	C_25_H_46_NO_7_P	503.3011	M+H	−0.10	461.2635	↑
P4	8.89	4-Oxoretinol	C_20_H_28_O_2_	300.2089	M+H	−0.16	185.1337, 129.0705, 119.0866, 105.0700	↑
P5	8.88	Perindoprilat	C_17_H_28_N_2_O_5_	340.1998	M+H	4.09		↑
P6	8.83	LysoPC (20:3(5Z,8Z,11Z)/0:0)	C_28_H_52_NO_7_P	545.3481	M+H	−0.23	546.3555, 528.3448, 469.2714, 363.2890	↑
P7	7.60	LysoPC (20:4(5Z,8Z,11Z,14Z)/0:0)	C_28_H_50_NO_7_P	543.3324	2M+H	0.93	526.3298	↑
P8	6.24	LysoPC (20:5(5Z,8Z,11Z,14Z,17Z)/0:0)	C_28_H_48_NO_7_P	541.3168	M+H	−0.36	542.3251, 524.3132, 184.0733, 166.0626, 104.1067, 86.0962	↑
P9	6.09	LysoPE (0:0/20:5(5Z,8Z,11Z,14Z,17Z))	C_25_H_42_NO_7_P	499.2698	M+H	−0.70	481.2322	↑
P10	5.89	Fosinopril	C_30_H_46_NO_7_P	563.3011	M+H	−4.11	564.3066	↑
P11	0.81	Imidazoleacetic acid ribotide	C_10_H_15_N_2_O_9_P	338.0515	M+H	−1.60	124.9995	↑
P12	10.91	12-HETE	C_20_H_32_O_3_	320.4663	M-H_2_O-H, M-H	0.11	319.2277, 301.2175, 257.2285, 179.1076	↑
P13	10.69	Medroxyprogesterone	C_22_H_32_O_3_	344.4877	M-H_2_O-H, M-H	−4.14	343.2291, 303.2329	↑
P14	9.00	Leukotriene A4	C_20_H_30_O_3_	318.4504	M-H_2_O-H, M-H	1.63	317.2131	↑
P15	7.59	LysoPE (0:0/20:4(5Z,8Z,11Z,14Z))	C_25_H_44_NO_7_P	501.5931	M-H, 2M-H	−0.20	500.2783, 303.2329, 259.2431, 214.0484, 205.1959, 196.0376, 140.0117	↑
P16	15.91	Retinyl ester	C_20_H_30_O_2_	302.451	M-H	−0.05	301.2174, 257.2295, 203.1808	↑
P17	16.87	Tetrahydrodeoxycorticosterone	C_21_H_34_O_3_	334.4929	M-H_2_O-H	0.39	315.2328, 149.0970	↑
P18	16.86	Hypogeic acid	C_16_H_30_O_2_	254.4082	M-H	−0.62	253.2171	↑
P19	16.27	13-HODE	C_18_H_32_O_3_	296.4449	M-H_2_O-H	−0.02	277.2172	↑
P20	16.66	PE (22:6(4Z,7Z,10Z,13Z,16Z,19Z)/18:0)	C_45_H_78_NO_8_P	792.0765	M-H	4.23	790.5402, 480.3099, 462.2997, 419.2538, 283.2625	↑
P21	16.53	PC (22:6(4Z,7Z,10Z,13Z,16Z,19Z)/TXB2)	C_50_H_82_NO_12_P	920.175	M-H	1.64	885.5507, 883.5359, 857.5224, 581.3108, 327.2328	↑
P22	13.38	Deoxycorticosterone	C_21_H_30_O_3_	330.4611	M-H_2_O-H	−0.18	311.2014, 149.0969	↑
P23	14.92	LysoPI (18:0/0:0)	C_27_H_53_O_12_P	600.6763	M-H	2.48	599.3208, 315.0488, 297.2800, 241.0117	↑
P24	13.52	LysoPI (0:0/18:0)	C_27_H_53_O_12_P	600.6763	M-H	1.88	599.321	↑
P25	17.71	Carbocyclic thromboxane A2	C_22_H_36_O_3_	348.2664	M-H_2_O-H	0.00	329.2480, 277.1810, 259.1692, 191.1813	↑
P26	17.35	Arachidonic acid	C_20_H_32_O_2_	304.2402	M-H	0.28	303.2328, 259.2430, 205.1961	↑
P27	17.06	PA (14:0/22:4(7Z,10Z,13Z,16Z))	C_39_H_69_O_8_P	696.947	M-H_2_O-H	0.59	677.4551, 438.2284, 347.2013, 345.1841, 191.1804	↑
P28	17.60	16-Dehydroprogesterone	C_21_H_28_O_2_	312.2089	M-H	0.11	311.2021, 149.0971	↑
P29	17.59	PA (14:1(9Z)/PGF1alpha)	C_37_H_67_O_11_P	718.442	M-H	−3.74	717.4325, 279.2330	↑
P30	17.58	DG (15:0/20:5(5Z,8Z,11Z,14Z,17Z)/0:0)	C_38_H_64_O_5_	600.4753	M-H_2_O-H	−2.59	581.4550, 301.2173, 279.2331, 241.2172	↑
P31	17.49	PGP (a-13:0/a-25:0)	C_44_H_88_O_13_P_2_	886.57	M-H_2_O-H	1.50	867.5537, 774.5740, 756.5626	↑
P32	7.34	LysoPE (24:6(6Z,9Z,12Z,15Z,18Z,21Z)/0:0)	C_29_H_48_NO_7_P	553.6677	M-H	0.74	552.3095, 283.2451, 229.1961, 152.9956	↑
P33	7.33	LysoPE (22:4(7Z,10Z,13Z,16Z)/0:0)	C_27_H_48_NO_7_P	529.6463	M-H	0.96	528.3100, 259.2430, 229.1964, 205.1963, 152.9955	↑
P34	7.18	PC (17:2(9Z,12Z)/0:0)	C_25_H_48_NO_7_P	505.633	M-H	0.03	504.3096, 279.2331, 224.0690	↑
P35	7.17	LysoPE (20:4(5Z,8Z,11Z,14Z)/0:0)	C_25_H_44_NO_7_P	501.5931	M-H	0.59	500.2784, 214.0485, 152.9958	↑
P36	7.67	PC (2:0/6 keto-PGF1alpha)	C_30_H_54_NO_12_P	651.731	M-H_2_O-H	−3.81	632.3188, 500.2043, 224.0690	↑
P37	0.83	N-Acetyltaurine	C_4_H_9_NO_4_S	167.18	M-H	−1.11	166.0176, 124.0071, 79.9575	↑
P38	6.33	LysoPE (22:5(4Z,7Z,10Z,13Z,16Z)/0:0)	C_27_H_46_NO_7_P	527.6304	M-H	0.93	526.2940, 327.2333, 257.2274, 203.1803, 152.9958	↑
P39	2.79	Tauroursocholic acid	C_26_H_45_NO_7_S	515.703	M-H	1.17	514.2848	↑
P40	2.54	Vanilpyruvic acid	C_10_H_10_O_5_	210.1834	M-H	−1.39	209.0453, 165.0556, 121.0292	↑
P41	11.94	LysoPC (17:0/0:0)	C_25_H_52_NO_7_P	509.6566	M-H	0.23	508.3407, 419.2569, 283.2644, 224.0692	↑
P42	11.60	LysoPE (18:0/0:0)	C_23_H_48_NO_7_P	481.6035	M-H	1.21	480.3100, 307.2635, 295.2645, 283.2635	↑
P43	10.64	Deoxycholyltyrosine	C_33_H_49_NO_6_	555.756	M-H	−3.47	494.3261, 492.3465	↑
P44	12.73	LysoPE (20:0/0:0)	C_25_H_52_NO_7_P	509.6566	M-H	−0.10	508.3408, 152.9956	↑
P45	12.36	LysoPE (0:0/18:0)	C_23_H_48_NO_7_P	481.6035	M-H	0.36	480.3099, 307.2628, 214.0480, 196.0377, 140.0112	↑
P46	11.95	Stearic acid	C_18_H_36_O_2_	284.4772	M-H	0.04	283.2645	↑
P47	8.44	LysoPI (20:4(5Z,8Z,11Z,14Z)/0:0)	C_29_H_49_O_12_P	620.6659	M-H	0.30	619.2889, 439.2256, 315.0487, 303.2328, 259.2430, 241.0115	↑
P48	1.26	Uric acid	C_5_H_4_N_4_O_3_	168.1103	M-H	−1.80	167.0208, 166.0132, 124.0146	↑
P49	17.93	PS (20:0/PGJ2)	C_46_H_80_NO_12_P	869.5418	M+H	−0.55	870.5510, 747.4920, 95.0852	↓
P50	17.88	PGP (18:0/PGF1alpha)	C_44_H_84_O_16_P_2_	930.5234	M+H	−2.18	931.5286	↓
P51	14.82	LysoPE (0:0/22:0)	C_27_H_56_NO_7_P	537.3794	M+H	−0.85		↓
P52	13.51	LysoPC (22:4(7Z,10Z,13Z,16Z)/0:0)	C_30_H_54_NO_7_P	571.3637	M+H	−4.14	572.3691, 513.2946, 104.1067	↓
P53	11.39	LysoPE (0:0/22:1(13Z))	C_27_H_54_NO_7_P	535.3637	M+H	−0.35		↓
P54	10.74	LysoPC (20:2(11Z,14Z)/0:0)	C_28_H_54_NO_7_P	547.3637	M+H	−0.85	530.3605, 441.2401, 166.0631, 104.1068, 86.0962, 60.0807	↓
P55	6.62	LysoPE (0:0/18:1(11Z))	C_23_H_46_NO_7_P	479.5876	M-H	0.79	478.2942	↓
P56	7.97	LysoPE (0:0/22:4(7Z,10Z,13Z,16Z))	C_27_H_48_NO_7_P	529.3168	M+H	−4.81	472.2565	↓
P57	7.45	Dodecanoylcarnitine	C_19_H_37_NO_4_	344.2795	M+H	1.55		↓
P58	6.69	LysoPC (16:1(9Z)/0:0)	C_24_H_48_NO_7_P	493.3168	M+H	−0.78	312.2635, 311.2581	↓
P59	16.32	PC (20:2(11Z,14Z)/TXB2)	C_48_H_86_NO_12_P	900.185	M-H	−1.37	854.5928, 790.5403, 305.2480	↓
P60	15.59	N-Oleoyl phenylalanine	C_27_H_43_NO_3_	429.645	2M-H	−0.74		↓
P61	13.67	LysoPE (22:1(13Z)/0:0)	C_27_H_54_NO_7_P	535.6939	M-H	1.16	534.3574, 391.2117, 152.9957	↓
P62	17.16	PE-NMe2 (18:0/20:4(8Z,11Z,14Z,17Z))	C_45_H_82_NO_8_P	796.124	M-H_2_O-H	−0.44	776.5612, 766.5397, 740.5261, 508.3412, 490.3309, 303.2328, 283.2624, 259.2429	↓
P63	17.66	PS (18:1(11Z)/6 keto-PGF1alpha)	C_44_H_78_NO_14_P	875.5159	M-H	0.95		↓
P64	10.23	Chenodeox–cholyltyrosine	C_33_H_49_NO_6_	555.756	M-H	−2.91	506.3260, 494.3263	↓
P65	8.76	LysoPC (15:0/0:0)	C_23_H_48_NO_7_P	481.6035	M-H	−0.47	480.3093, 255.2332, 224.0693	↓
P66	9.79	LysoPE (20:1(11Z)/0:0)	C_25_H_50_NO_7_P	507.6408	M-H	−0.29	506.3250, 152.9956	↓
P67	9.17	PC (PGF2alpha/2:0)	C_30_H_54_NO_11_P	635.732	M-H	−2.44		↓

Abbreviations: “↑”—Compared with the normal group, the relative content of compounds in the model group was up-regulated; “↓”—Compared with the normal group, the relative content of compounds in the model group was down-regulated.

**Table 3 nutrients-14-03926-t003:** Alpha diversity index analysis.

Sample	Control	Model	FG	GG
Chao 1	456.22 ± 16.934	442.16 ± 30.687	448.97 ± 19.631	457.52 ± 53.739
Ace	447.01 ± 12.98	439.96 ± 30.287	439.38 ± 17.572	453.51 ± 61.46
Shannon	4.4543 ± 0.13655	4.4905 ± 0.090045	4.6394 ± 0.094192 **	4.595 ± 0.098015 *
Simpson	0.02326 ± 0.0049968	0.021155 ± 0.0021977	0.017352 ± 0.0030513 **	0.018555 ± 0.0026981 *
Coverage	0.99875 ± 0.00011005	0.99878 ± 0.00019433	0.99883 ± 0.00013631	0.99873 ± 0.00048036

*n* = 10; * *p* < 0.05; ** *p* < 0.01 compared with the model group.

## Data Availability

The data used to support the findings of this study are available from the corresponding author upon request.

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
