# Peer review of "Efficacy and Mechanism of Pueraria lobata and Pueraria thomsonii Polysaccharides in the Treatment of Type 2 Diabetes"

_nutrients, 2022, doi:10.3390/nu14193926_

Round 1
Reviewer 1 Report
The authors combined the metabolomics approach with several other approaches to show the antidiabetic activity and mechanism of two species: Pueraria lobata and thomsonii polysaccharides in diabetic rats.
The study is well designed. However, I have a few concerns as below:
1- The title can be shorter. I do not think you need to mention two times polysaccharides.
2- In section 2,3,2 LC-MS: I do not see the model and specificity of the mass spectrometer. Please mention the exact model of the LC and MS for more producibility.
3- I wonder, since the authors used a qTOF system for the identification of metabolites (from HMDB), why did they not try to identify the metabolites in plants? There are several public libraries, such as MoNA (Mass Bank of North America) and RIKEN (http://prime.psc.riken.jp/compms/msdial/main.html) which provide the MSMS spectra for peak annotation. Please clarify why metabolites in plants were not subjected for identification.
4- In Figures 7,15, and 16, the text is not readable.
5- Authors used the OPLS-DA model, which requires to be validated. Please provide the power of the analysis (R2 and Q2 values).
Reviewer 2 Report
The article "Efficacy and mechanism of Pueraria lobata polysaccharide and Pueraria thomsonii polysaccharide in the treatment of type 2 diabetes" offers a thorough analysis of the effects of the two chosen plants. However, major drawback of the article are the claims connected to specific phytochemical constituents and their presumed biological effects. In order to claim that the polysaccharides are responsible for the observed action, their type and quantity in plant material should be established by (phyto)chemical methods.
Some other, minor comments are as follows:
Please write binomial plant names in italic and provide the name of the botanical authorities and plant family on the first mention (e.g. line 57 and elsewhere).
If abbreviation for plant names was inroduced, then the full names of the plants should be mentioned only once,. and only abbreviations used from then on.
All the abbreviations used in the tables should be explained, either in the footnote or in the table title (e.g. table 2 and elsewhere).
Round 2
Reviewer 2 Report
No further comments.